# PETNet– Coincident Particle Event Detection using Spiking Neural Networks

## Abstract

Spiking neural networks (SNN) hold the promise of being a more biologically plausible, low-energy alternative to conventional artificial neural networks. Their time-variant nature makes them particularly suitable for processing time-resolved, sparse binary data. In this paper, we investigate the potential of leveraging SNNs for the detection of photon coincidences in positron emission tomography (PET) data. PET is a medical imaging technique based on injecting a patient with a radioactive tracer and detecting the emitted photons. One central post-processing task for inferring an image of the tracer distribution is the filtering of invalid hits occurring due to e.g. absorption or scattering processes. Our approach, coined PETNet, interprets the detector hits as a binary-valued spike train and learns to identify photon coincidence pairs in a supervised manner. We introduce a dedicated multi-objective loss function and demonstrate the effects of explicitly modeling the detector geometry on simulation data for two use-cases. Our results show that PETNet can outperform the state-of-the-art classical algorithm with a maximal coincidence detection $F_1$ of 95.2%. At the same time, PETNet is able to predict photon coincidences up to 36 times faster than the classical approach, highlighting the great potential of SNNs in particle physics applications.

## 1 Introduction

In recent years spiking neural networks (SNN) have received increasing research interest, given their potential as a low-energy and computationally efficient alternative to classical artificial neural networks (ANNs). Spiking neurons model their activations as time-dependent membrane potentials, thereby mimicking brain functionality more closely. Information is propagated via binary spike trains, naturally capturing the sparsity and temporal dynamics found in their biological analogue Roy et al. (2019). The intrinsic concept of sparsity and temporal dynamics enables SNNs to model sparse, time-resolved data, which can be found in a wide variety of scientific applications. Especially for high-frequency data, conventional ANNs struggle with the additional computational complexity introduced by the temporal dimension. Recurrent ANN architectures, such as LSTMs, are not able to capture long range dependencies, while Transformer-based approaches face the challenge of excessive memory consumption for long sequences. For such cases, SNNs provide a promising alternative with the potential of surpassing conventional ANN prediction accuracy while requiring substantially less computational resources.

One such potential application is the task of detecting photon coincidence pairs from raw detector data in positron emission tomography (PET). PET is a medical imaging technique based on injecting a radioactive positron-emitter into a patient and measuring the emitted photons of the resulting electron-positron-annihilation using a scintillating ring detector. In theory coincidence pairing can be performed by simply matching two measured signals within a predefined temporal window. In practice, however, not all annihilation photons can be detected due to absorption processes in the body or physical constraints of the scintillator crystals, thus impeding the matching process. At high signal rates and temporal resolutions, existing classical algorithms prevalent in the PET-community reach their limits as computation times can require several hours. We demonstrate that SNNs can speed up this task and even improve detection accuracy by learning coincidence patterns. The photons detected by the scanner inherently comprise a time-variant series of binary events in discrete locations, and thus can be easily translated into spike-trains interpretable by an SNN.

Our contributions are as follows:

- A multi-objective loss function for SNNs that is both sensitive to spike counts and timing critical in particle detection applications.

- An approach to model the a-priori known detector geometry with window functions.

- Large-scale data-parallel SNN training on a multi-node GPU systems for extremely high-rate PET scanners.

- Prediction of photon coincidence pairs in PET using SNNs, on-par or even surpassing the classical approach in prediction accuracy while reducing inference time by more than an order of magnitude.

We evaluate our approach on two sets of simulations, one based on a high-rate preclinical system and one based on a low-rate monolithic clinical design, which will be made publicly available to serve as benchmark datasets. Our work provides a real-world use-case for the application of SNNs, yielding fast and precise detection of coincident particle events.

## 2 BACKGROUND

### 2.1 POSITRON EMISSION TOMOGRAPHY

PET is based on attaching a radioactive $\beta^+$-emitter to a metabolic substance, e.g. sugar, and injecting this so-called radiotracer into the subject of study. The tracer accumulates in certain tissues according to its metabolic properties, where it undergoes a $\beta^+$-decay. The emitted positrons annihilate with electrons in the surrounding tissue, resulting in two $511\,\mathrm{keV}$ back-to-back photons which are measured in a scintillating ring-detector. By connecting coincident pairs of detected photons as so-called lines of response (LOR), an image can be reconstructed from the entirety of all LORs, i.e. the sinogram, as an inverse problem.

Since radioactive decay is a stochastic process, measurements are conducted over a time-span to acquire sufficient signal. In a clinical setting, the goal is to keep the radioactive dose to the patient as low as possible. Therefore, long measurement times at low activities are used to gather sufficient statistics. However, these low-activity measurements do not allow the observation of fast kinematic processes that happen on the seconds-to-minutes scale and are usually completed within a fraction of the acquisition-time of a PET image. For research purposes, preclinical studies are conducted on mice and rats, allowing for much higher radiation doses and faster acquisition times. For example, the SAFIR-project Ritzer et al. (2020) provides a detector capable of processing vastly increased measurement activities, shortening the image acquisition time from several minutes to less than 10 seconds.

PET image reconstruction is based on sorting the individually detected photons into coincidence pairs. However, a significant fraction of photons is lost due physical effects such as the limited detection efficiency and time-resolution of the detector, as well as photons being subject to photo-electric absorption processes in the body. Hence, a lot of single events are registered, only a fraction of which has a detected coincidence partner. A major preprocessing step in PET image reconstruction is thus filtering out all coincidence pairs from the entirety of detector hits.

To date, coincidence pair matching is still performed using non-ML approaches. Common techniques rely on storing all detected photons as a list of 'single' events with a given energy, time and location (singles list-mode data), and iteratively filtering this list post-acquisition Oliver et al. (2008). The *Single-Coincidence-Window-sorting (SCW)* algorithm, a commonly used variant, sequentially processes such a time-sorted list of single events using a fixed time 'window' dependent on the detectors timing resolution. For a given event, the algorithm examines if the time-difference to the next single event is smaller than the window. If not, the Type I event is rejected as a 'single', and the algorithm moves on to the next event in the list. If yes, then both are accepted as coincident Type II event and translated into a line-of-response (LOR). Should more than one event fit inside the 'time-window', all are rejected (Type III), as it cannot be known which two hits among the multiple correspond to a LOR. Conventionally, the SCW algorithm is additionally supplied with geometrical information, rejecting highly unlikely coincidences such as those between neighboring crystals. A

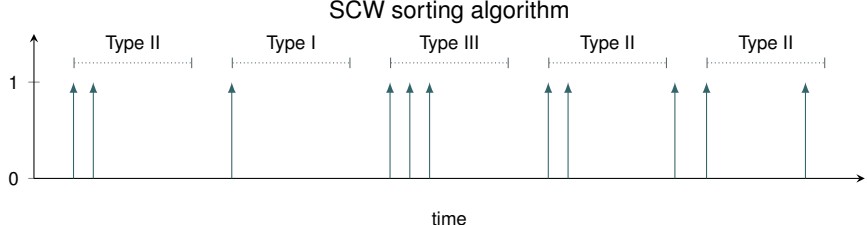

Figure 1: Schematic representation of the SCW sorting algorithm, modified from Oliver et al. (2008). Three event classes are displayed: (Type II) are two hits within the defined coincidence window frame (gray dotted), resulting in an accepted coincidence. (Type I) is rejected as only one hit is detected, (Type III) is rejected as more than two hits are registered.

schematic detailing the algorithm is shown in Figure 1. Due to its sequential nature, the SCW algorithm scales heavily with the signal acquisition rate, a property that becomes especially problematic at high frequency measurements as those encountered in preclinical PET.

## 2.2 LEAKY INTEGRATE FIRE SPIKING NEURAL NETWORKS

Spiking neural networks try to mimic information propagation as closely as possible to the biological analog of the brain Roy et al. (2019). The most prominent difference to ANNs is the integration of the temporal component, i.e. the activation of a spiking neuron is time dependent, and spike propagation in a network takes place asynchronously. The *Leaky Integrate and Fire (LIF)* neuron is a common mathematical approach to model neural activity in spiking neural networks Lapicque (1907); Yamazaki et al. (2022). Formally, LIFs can be described recursively as follows:

$$M_{t+1} = W X_{t+1} + \alpha M_t - \alpha S_t \quad \text{with} \quad S_t = \begin{cases} 1, & \text{if } M_t > M_{threshold} \\ 0, & \text{otherwise} \end{cases} \quad (1)$$

where $t$ denotes the discretized time step of the input spike train $X$, $M$ the membrane potential, $M_{threshold}$ the maximum membrane capacity, $S$ the output spike, $W$ a trainable weight matrix and $\alpha$ a decay value for the potentials. $\alpha$ is usually a constant, scalar hyperparameter to the model in the range $[0.9, 0.99]$, but may also be learned and dynamically inferred Ding et al. (2022).

The above formalism allows the interpretation of a spiking neural network with LIF neurons as a recurrent neural network and by extension enables gradient-based training with backpropagation through time Lee et al. (2016); Wu et al. (2018). To overcome the *dead neuron phenomenon*, resulting from the non-differentiability of the Heaviside spiking function, *surrogate gradient* techniques can be employed during training Neftci et al. (2019). In particular, the Heaviside gradient is smoothed out and replaced with the $arctan$ function to enable training with widely used stochastic gradient descent approaches as follows:

$$\frac{\partial S}{\partial M} = \frac{1}{\pi(1 + [M\pi]^2)}. \quad (2)$$

## 3 SPIKING NEURAL NETWORKS FOR PET

### 3.1 PROBLEM STATEMENT

We formulate coincidence pairing as a supervised learning problem in which a set of binary time series for each crystal is denoised and time-normalized to only identify Type II events Bellec et al. (2018); Ciurletti et al. (2021). A schematic overview of this process is depicted in Figure 2. More formally, we consider the hits to be a sparse binary event spike train $X \in \{0, 1\}^{C \times T}$ where $C$ is the crystal count and $T$ the number of discretized time steps. Based on this input spike train, we want to predict an output spike train $S \in \{0, 1\}^{C \times T}$ for a given $X$ as close as possible to a label

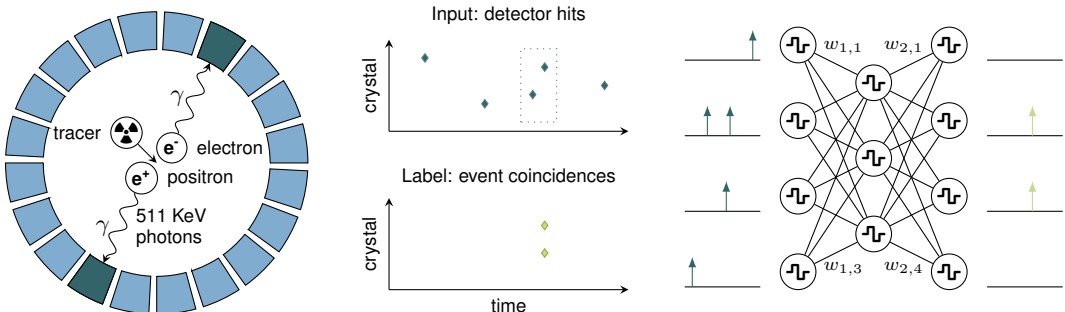

Figure 2: Overview of PETNet's coincidence detection process. Left: functional principle of PET – detection of photons emitted through annihilation of the $\beta^+$ tracer particles with tissue electrons. Center: input spike trains of detector hits and corresponding coincidences (gray dotted box). Right: PETNet, a supervised denoising spiking neural network with LIF neurons.

$Y \in \{0,1\}^{C \times T}$. $Y$ is 1 if and only if two crystals $c_1$ and $c_2$ have a coinciding photon hit for the latter of the two time-steps. Note that $c_1$ and $c_2$ will often be spatially opposite of one another, but do not necessarily have to be. In the following we will generally use the subscripts $c$ and $t$ to denote a crystal or time-step of $X$, $S$ and $Y$ respectively.

## 3.2 MULTI-OBJECTIVE LOSS FUNCTION

Naïvely, it is reasonable to assume that we can model the above stated problem by minimizing the crystal- and time-step-wise binary cross-entropy loss between the network's prediction $S$ and the targets $Y$. Practically, this is infeasible and will not lead to convergence due to the label sparsity. To facilitate model training, we require a loss metric that incentivizes the correct number of output spikes per crystal, while also ensuring a correct arrival time of said spikes. Hence, we propose the $L_{PETNet}$ loss function as the weighted sum of two separate, but not independent factors as follows for a spike-train-label-pair $S, Y$:

$$L_{PETNet}(S, Y) = aL_\Gamma(S, Y) + bL_\Delta(S, Y), \tag{3}$$

where $a$ and $b$ are scalar hyperparameters weighting the individual loss contributions. To explain both terms $L_\Gamma$ and $L_\Delta$ in details, we need to introduce a helper function that identifies the set of spike timing indices for a given spike train:

$$I_1(X) = \{t | \forall x(t) \in X : x(t) = 1\} \tag{4}$$

For easier readability, we will define $I_S = I_1(S)$ and $I_Y = I_1(Y)$ to be sets of spike timing indices of the prediction and the labels respectively. Then, the spike count loss $L_\Gamma$ is the sum of all crystal-wise mean-squared-errors of the cardinalities of the spike index sets:

$$L_\Gamma(S, Y) = \sum_{c=1}^{C} MSE\left(|I_{S,c}|, |I_{Y,c}|\right) = \frac{1}{C} \sum_{c=1}^{C} (|I_{S,c}| - |I_{Y,c}|)^2 \tag{5}$$

The timing loss $L_\Delta$ can be expressed in a multitude of ways. Two possible choices are either the crystal-wise mean-squared error or, alternatively, the Chamfer distance Fan et al. (2017) of the predicted and labeled spike timings. In other words, this loss component measures the average number of time steps between an output spike and its next closest label spike and additionally the inverse direction for the Chamfer loss:

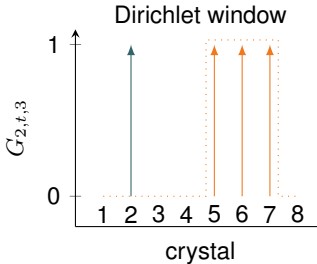 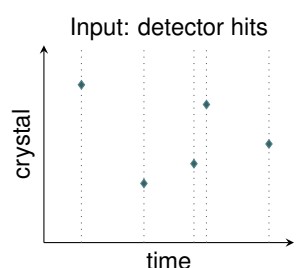 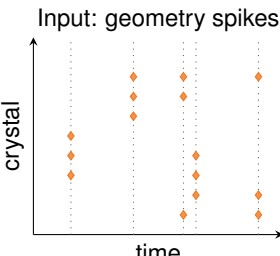

Figure 3: Visualization of the explicit geometry modeling. Left: Dirichlet window (orange) for a detector with $C = 8$ crystals, a hit at crystal $c = 2$ (blue) and window size $w = 1$. Center: time-resolved detector hits. Right: corresponding time-resolved geometry spikes.

$$L_{\Delta,MSE}(S,Y) = \sum_{c=1}^{C} MSE\left(I_{S,c}, I_{Y,c}\right) = \sum_{c=1}^{C}\left(\sum_{s \in I_{S,c}} \min_{y \in I_{Y,c}} \|s - y\|_2^2\right) \tag{6}$$

$$L_{\Delta,Chamfer}(S,Y) = \sum_{c=1}^{C}\left(\sum_{s \in I_{S,c}} \min_{y \in I_{Y,c}} \|s - y\|_2^2 + \sum_{y \in I_{Y,c}} \min_{s \in I_{S,c}} \|y - s\|_2^2\right) \tag{7}$$

### 3.3 MODELING DETECTOR GEOMETRY

The geometrical arrangement of scintillator crystals in a PET scanner is essential in detecting co-incidence pairs. In principle a spiking neural network should be able to infer said geometry automatically given enough training data. Yet, we may shorten training times and possibly obtain better predictive performance if we pass this a priori knowledge explicitly into the network as additional input. One possible modeling approach is to introduce an additional geometric feature for every crystal that spikes if the crystal opposite has detected a particle hit. To account for non-centered tracers, close-by crystals may equally spike in a parametric distance $w$. A graphical representation is depicted in Figure 3. More formally, we can define a geometry spiking function $G$ by creating a Dirichlet window Fessler & Sutton (2003) as follows:

$$G_{c,t,w} = \begin{cases} 1, & \text{if } \exists i \in \{-w, ..., w\} : S_{mod(c+i+C/2,C),t} = 1 \\ 0, & \text{otherwise,} \end{cases} \tag{8}$$

where $i$ signifies the index of the opposite crystals and all of them are arranged ascending in a ring. Then, the extended input $X^*$ to the spiking neural network becomes:

$$X^* = \begin{pmatrix} X \\ G_{c,t,w} \circ X \end{pmatrix} \in \{0,1\}^{2C \times T}. \tag{9}$$

## 4 EXPERIMENTAL EVALUATION

### 4.1 DATASETS

We investigate the feasibility of SNNs particle coincidence detection on simulation data with known ground truth. We consider two application use cases. The **Clinical** dataset models a setting with a low number of $C = 240$ crystals and low tracer activity, resulting in a small number of input spikes. This configuration is common in clinical monolithic crystal PET scanners such as Mendes et al. (2008). The **SAFIR** dataset represents a pre-clinical setting with a large number of $C = 2880$ crystals and high activity akin to Ritzer et al. (2020). In both cases, we decompose a singular long

Table 1: Hyperparameters and training configuration of the SNN. $n_{epochs}$ is the number of epochs until early stopping activated. $L_\Gamma$ is the count loss and $L_\Delta$ are timing losses, i.e. $L_{\Delta,MSE}$ and $L_{\Delta,Chamfer}$. $B$ is the local batch size per GPU.

| Dataset | $n_{in}$ | Layers | $n_{hidden}$ | $lr$ | $n_{epochs}$ | | $B$ | GPUs |
|---|---|---|---|---|---|---|---|---|
| | | | | | $L_\Gamma$ | $L_\Gamma + L_\Delta$ | | |
| **Clinical** | 240 | 1 | 368 | 2.454e-3 | 19 | 31 | 64 | 8 |
| with geometry | 480 | 1 | 368 | 2.454e-3 | 18 | 11 | 64 | 8 |
| **SAFIR** | 2880 | 1 | 4416 | 2.454e-3 | - | 26 | 8 | 64 |
| with geometry | 5760 | 1 | 4416 | 2.454e-3 | - | 12 | 8 | 64 |

time series of detector hits, as it would occur in practice, into a large set of shorter disjoint spike-trains for training, that spans $T = 2000$ discrete time steps in arbitrary units. The data was generated in a Monte Carlo fashion akin to GATE algorithms Jan et al. (2004) as follows: 1. a discrete number of decay-events is uniformly randomly distributed over the total number of time-steps based on the tracer activity, 2. for each event two target positions are designated at random, with the only constraint being that they are spatially opposite each other (with a potential shift of $\pm 2$ crystals to emulate a limited spatial resolution), and 3. spikes are generated at each of these positions with a set probability; if both target positions for a given event generate a spike, the target event and its corresponding positions will be marked in the label data. Hence, every sample consists of inputs $X$ and targets $Y$ with dimensions $C \times T$, where $X$ contains binary encoded detector hits in crystals $c \in C$ at time $t$, and $Y$ contains one-hot encoded coincidence photons in crystals $c' \in c$ at time $t$. To investigate the impact of using the detector geometry as additional input feature, each sample was enriched by the geometric features as described in Section 3.3. In this configuration, the second half of the input neurons is utilized to provide geometric information to the net, highlighting positions opposite of the original event to emphasize potential coincidence-pairs. For each of the two use cases, a total number of 60 000 samples was generated, both with and without detector geometric features. We further generated a reduced dataset, consisting of 8 000 samples, generated in the low-activity clinical PET scanner setting with only $T = 1000$ time steps. This dataset was utilized for initial testing and hyperparameter optimization, since it requires less computational resources and shorter training times compared to the full dataset. All data is publicly available and may be freely used for reproduction or related experiments[1].

## 4.2 SNN Architecture

We use a standard fully-connected SNN model architecture with $C$ input and output neurons and $n_{layer}$ hidden layers of size $n_{hidden}$. For the dataset with additional geometric features, the number of input neurons is $2C$. We trained the individual weights as well as the decay constant $\alpha$ for each LIF-neuron. We used 50 000 samples of each dataset for training, with an 80:20 split for training and validation data, while keeping the remaining 10 000 samples as a hold-out test-set for our final performance evaluation.

Optimization of the hyperparameters $n_{layer}$, $n_{hidden}$ and the learning rate $lr$ was conducted on the reduced dataset using the parallel genetic algorithm `propulate` Taubert et al. (2023). Since evaluation of the population individuals of the optimization indicated no correlation between batch size $B$ and predictive performance, we set the batch size to the maximum number of samples to fit on a single GPU. Hyperparameter optimization is a time consuming task, requiring immense amounts of compute and energy Gutiérrez Hermosillo Muriedas et al. (2023). Therefore we refrained from optimizing hyperparameters for the SAFIR dataset and instead, scaled the number of hidden neurons according to the same fraction with respect to the number of crystals determined for the clinical dataset, while keeping the number of hidden layers and the learning rate identical.

We trained the SNN model using the different loss functions described in Section 3.2, with weighting factors $a = 1$, $b = 0.1$, using the Adam optimizer Kingma & Ba (2015) and the identified training configuration listed in Table 1. Due to the large data size, we utilize data-parallel training to speed-up training and enabling feasible runtimes Coquelin et al. (2022). We employ early stopping on the

---

[1]Will be released upon publication

Table 2: Performance comparison of PETNet trained with different loss functions on the clinical PET scanner dataset with the traditional SCW approaches. The top values correspond to training runs including geometric features, the bottom values were achieved without geometric features.

| | Method | TP | FP | FN | $F_1$ | Precision | Time[min] |
|---|---|---|---|---|---|---|---|
| incl. geo. | SCW | 7618 | 828 | 266 | 0.933 | 0.902 | 63.91 |
| | SNN $L_\Gamma$ | 7032 | 1083 | 864 | 0.878 | 0.867 | 3.67 |
| | SNN $L_\Gamma + 0.1L_{\Delta,MSE}$ | 7324 | 595 | 572 | 0.926 | 0.925 | 3.70 |
| | SNN $L_\Gamma + 0.1L_{\Delta,Chamfer}$ | 7324 | 595 | 572 | 0.926 | 0.925 | 3.67 |
| w/o geo. | SNN $L_\Gamma$ | 6718 | 1575 | 1180 | 0.830 | 0.810 | 3.53 |
| | SNN $L_\Gamma + 0.1L_{\Delta,MSE}$ | 7576 | 439 | 322 | **0.952** | 0.945 | **3.45** |
| | SNN $L_\Gamma + 0.1L_{\Delta,Chamfer}$ | 7576 | 439 | 322 | **0.952** | 0.945 | 3.47 |

$F_1$ score with a patience of 3 for a maximum number of 50 epochs. $F_1$ was chosen as a stopping criterion instead of the validation loss because we observed a double descent behavior Nakkiran et al. (2021) in the loss, attributed to the two factor nature of our combined loss functions.

## 4.3 COMPUTATIONAL ENVIRONMENT

We ran all experiments on a distributed-memory, parallel hybrid supercomputer. Each of the compute nodes is equipped with two 38-core Intel Xeon Platinum 8368 processors at 2.4 GHz base and 3.4 GHz maximum turbo frequency, 512 GB local memory, a local 960 GB NVMe SSD disk, two network adapters and four NVIDIA A100-40 GPUs with 40 GB memory connected via NVLink. Inter-node communication uses a low-latency, non-blocking NVIDIA Mellanox InfiniBand 4X HDR interconnect with 200 Gbit/s per port. All experiments have been implemented in Python 3.9.1, `PyTorch` 2.0.1 Paszke et al. (2019) using `CUDA` 11.8 and `snnTorch` 0.7.0 Eshraghian et al. (2021). For data-parallel training, we utilized PyTorch's `DistributedDataParallel` class with `OpenMPI` 4.1. The source code for the implementation is publicly available[2].

## 4.4 METRICS

We evaluate PETNet's predictive performance on common classification metrics for the identification of coincidence pairs. Due to the high class imbalance, we are particularly interested in the true-positive ($TP$), false-positive ($FP$) and false-negative ($FN$) rates or, in other words, the fraction of correctly detected coincidence spikes in the prediction, the number of faulty additional spikes without matches in the label and the fraction of undetected spikes respectively. Using these values we also compute the $F_1$ scores as the harmonic mean of precision ($TP/(TP + FP)$) and recall ($TP/(TP + FN)$). For all rates we allow a possible timing delay of $\pm40$ time steps. Furthermore, we evaluate the inference time required to predict coincidence hits.

## 4.5 RESULTS

Table 2 shows prediction metrics and inference runtimes of the SNN trained on different loss functions in comparison to the classical SCW algorithm, evaluated on 10 000 samples of the hold out test set. SNNs were run on a single NVIDIA A100 GPU, while the SCW algorithm ran on two 38-core Intel Xeon Platinum 8368 processors. We further trained a single layer LSTM as a neural network baseline for the task of coincidence hit prediction, both with and without additional geometric features. However, the model did not converge, resulting in predicting no coincidence hits at all, i.e. $TP = 0$. We hypothesize that this is attributed to the large number of time steps and sparse input signal the recurrent LSTM needs to loop over.

A number of interesting observations can be made. For one, $L_\Gamma + L_{\Delta,MSE}$ and $L_\Gamma + L_{\Delta,Chamfer}$ yield superior prediction metrics compared to $L_\Gamma$, underlining the importance of the proposed multi-objective loss function. Moreover using either $L_{\Delta,MSE}$ or $L_{\Delta,Chamfer}$ to account for timing yields the exact same prediction values. However, using $L_{\Delta,Chamfer}$ resulted in three times longer train-

---

[2]Will be released upon publication

Table 3: Performance comparison of PETNet trained with different loss functions on the SAFIR dataset with the traditional SCW approaches. The top values correspond to training runs including geometric features, the bottom values were achieved without geometric features.

| | Method | TP | FP | FN | $F_1$ | Precision | Time [min] |
|---|---|---|---|---|---|---|---|
| incl. geo. | SCW | 1776 | 128 | 264 | **0.901** | 0.933 | 880.41 |
| | SNN $L_\Delta + 0.1 L_{\Gamma, MSE}$ | 1712 | 277 | 336 | 0.848 | 0.861 | 36.70 |
| | SNN $L_\Delta + 0.1 L_{\Gamma, Chamfer}$ | 1712 | 277 | 336 | 0.848 | 0.861 | 36.19 |
| w/o geo. | SNN $L_\Delta + 0.1 L_{\Gamma, MSE}$ | 1675 | 232 | 373 | 0.847 | 0.878 | **30.64** |
| | SNN $L_\Delta + 0.1 L_{\Gamma, Chamfer}$ | 1675 | 232 | 373 | 0.847 | 0.878 | 30.78 |

ing times compared to $L_{\Delta, MSE}$. We conclude that the second summation term in eq. (7) can be neglected and that there exists a symmetry between prediction and target. We further observe that adding the detector geometry as a feature improves prediction accuracy for $L_\Gamma$, but not for $L_\Gamma + L_{\Delta, MSE}$ and $L_\Gamma + L_{\Delta, Chamfer}$. In fact it slightly worsen predictive performance, but speeds up-convergence in terms of number of epochs until early stopping activates, c.f. Table 1. Our most important finding, however, is that SNNs trained on loss functions that account for timing are able to outperform the classical algorithm coincidence detection, while also computing approximately 20 times faster.

Given that our multi-objective loss function with timing outperforms using simply count loss $L_\Gamma$ w.r.t. prediction accuracy, we train an SNN on the SAFIR dataset using only the combined loss functions $L_\Gamma + L_{\Delta, MSE}$ and $L_\Gamma + L_{\Delta, Chamfer}$ and compare results to the classical approach. Prediction metrics and inference runtime, evaluated on the $10\,000$ sample hold out test set, are listed in Table 3. Again, $L_{\Delta, MSE}$ and $L_{\Delta, Chamfer}$ yield the exact same results. Using the geometry as an additional feature marginally improves prediction accuracy, while reducing the number of epochs to convergence, thus yielding faster training. Unlike for the clinical dataset, SNN prediction accuracy is not able to beat the classical SCW algorithm. We hypothesize that this is caused by 1. non-optimal hyperparameters of the model, since we only adapted those optimized on the clinical dataset; and 2. a lower number of training samples compared to the number of trainable parameters in the model. However, the difference in inference time becomes even more pronounced on this larger dataset, with the SCW algorithm taking $\approx 36$ times longer to evaluate $10\,000$ samples.

## 5 RELATED WORK

This section reviews existing ANN/deep-learning methods used in the field of PET, and provides a brief overview of SNN methods relevant to our work.

### 5.1 NEURAL NETWORKS IN PET

PET imaging consist of an intricate multi-step process, many parts of which have recently been revisited under the lens of deep-learning methods. Many existing approaches center around utilizing ANNs for the reconstruction of PET images as an alternative to existing iterative algorithms Pain et al. (2022). To name a few examples, Dong et al. (2020) address the topic of PET-imaging artifacts caused by photon attenuation via utilizing a deep-learning based method capable of converting non-corrected images to corrected ones. Rahman et al. (2022) demonstrate a method to track the intrinsic dose distribution during proton-therapy using a cGAN-based deep learning framework. Hamdi et al. (2022) utilize a CNN as a computer-aided-diagnostics tool to classify brain images of potential Alzheimer's patients post image acquisition. Leroux et al. (2009) investigate the low-level intrinsics of PET-instrumentation, implementing an ANN on a Field-Programmable-Gate-Array (FPGA) to determine the arrival time of gamma-rays with increased accuracy. Arabi et al. (2021) provide an extensive overview of various further topics in PET that have been addressed using AI, among them improvements in the energy- and timing-resolution of the detector front-end and de-noising approaches for low-dose data acquisitions. Notably, the field lacks neural approaches addressing the problem of coincidence sorting with Fuster-Garcia et al. (2010) being the only exception. They compare a single-hidden-layer fully-connected neural network with existing PET-coincidence search

techniques. In contrast to our work, however, the network was used to classify pairs of single events as coincident or not on a case by case basis similar to existing classical approaches. The detection efficiency is significantly lower and a continuous prediction mode like PETNet's SNN is not possible.

## 5.2 Spiking Neural Networks

Spiking neural networks are inherently time-variant, and the various forms of spiking neuron models are sensitive to coincident spike events by design. Kamaruzaman et al. (2015) investigate this behavior for the spike-response neuron model, and present an application with regards to face recognition. This property of SNN's makes their application a potential candidate as a level-two trigger such as those employed in particle detection experiments, as R. Kulkarni et al. (2023) details as well. There, SNN's have been studied in relation to their ability to detect high-$p_T$ events at LHC, a related approach to PETNet's attempt to detect coincident particle events in PET.

A key concept in current applications of SNNs is that of spike-encoding. Existing software packages such as `snntorch` Eshraghian et al. (2021) differentiate between three separate methods; rate-encoding, in which input features are represented as spike frequencies, latency-encoding, in which input features determine the arrival time of individual spikes, and delta-modulation, where spikes indicate the temporal change of input features. As a result, common loss functions target one or several aspects of these encodings. Examples for classification tasks include cross-entropy-spike-count loss, encouraging the correct class to fire at all times, while suppressing all others. Equally, cross-entropy-temporal loss, which similarly encourages the correct class to fire before all others. Alternatively, losses such as the mean-squared-error-membrane loss target the output layer's membrane potentials instead of the spike count. Recently, more sophisticated methods such as the Information Maximization Loss presented by Guo et al. (2022) aim to improve SNNs with insights into Information theory.

## 6 Conclusion

In this work we present PETNet, a spiking neural network architecture for coincidence photon detection in positron emission tomography. PETNet introduces a novel multi-objective loss function which accounts for spike counts and timing jointly. It further provides a method to model the a-priori known detector geometry using window functions. Our evaluations on simulation data from two application use-cases demonstrate that PETNet can yield a prediction accuracy surpassing the state-of-the-art analytical approach while drastically reducing computation time.

Our approach provides a proof of concept with promising initial results, indicating that utilizing SNNs for coincidence detection in PET is a path worth pursuing. On a broader scale, our work highlights the potential of SNNs to be used as efficient level two trigger mechanisms such as those utilized in various particle detectors, with coincidence detection in PET being merely a specialized example. Future research could focus on many more such applications across the field of particle physics. One advantage worth noting is that the SNN approach allows deployment for real time prediction during scan time. The potential deployment of SNNs on low-power neuromorphic hardware Kösters et al. (2023) enables integration of the algorithm directly into the detector ring for on-board prediction with minimal computational requirements. This could drastically reduce data transfer between the scanner and the offline compute infrastructure, since only registered coincidences need to be transmitted. Moreover, the online prediction can be used for iterative image reconstruction, and thus the option to stop the scan prematurely when sufficient image quality is reached.

Our study reveals current challenges in real world applications of SNNs. Although SNNs utilize, in theory, sparse computations, current software implementations still operate on dense matrix-matrix multiplication, making the algorithm vastly inefficient and causing the need for large scale computational resources. As this goes against the overarching promise of SNNs being highly energy-efficient, future work requires truly sparse implementations of tensor computations. Nonetheless, our results demonstrate the great potential of SNNs for efficient modeling of sparse, binary, time-resolved data.

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
