# SUPPLEMENTARY MATERIALS

## 1 EXTENDED EVALUATION: COMPARISON TO LSTM

To compare our PETNet SNN to an ANN architecture, we trained an LSTM on the Clinical dataset for the task of coincidence pair prediction. The LSTM contained a single hidden layer with 240 nodes. We used the same settings as for the SNN, for 30 epochs on MSE spiking loss ($L_\Gamma$) using the Adam optimizer with a learning rate of $lr = 2.454 \times 10^{-3}$. We trained the LSTM in a similar data-parallel fashion on eight A100 GPUs with a local batch size of 64 samples per GPU. However, even though the loss converged (c.f. Figure 1), the model failed to predict any coincidence pairs. Both $F_1$ score and $Precision$ remained at 0.0 throughout the entire training. Looking at the course of $TP$, $FP$ and $FN$ compared to the number of true coincidences in Figure 3, it becomes clear that the model simply learns to set more and more output values to zero, naïvely reducing the loss by driving into a local minimum, unable to extract meaningful patterns. We observed similar behavior for binary cross entropy (BCE) loss and the combined loss with $L_\Delta$, as well as for an LSTM network with two hidden layers.

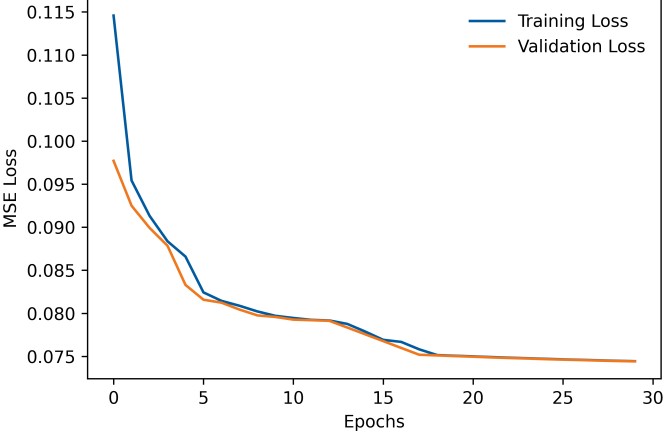

Figure 1: Training and validation loss of a single layer LSTM trained using the MSE loss ($L_\Gamma$) for 30 epochs.

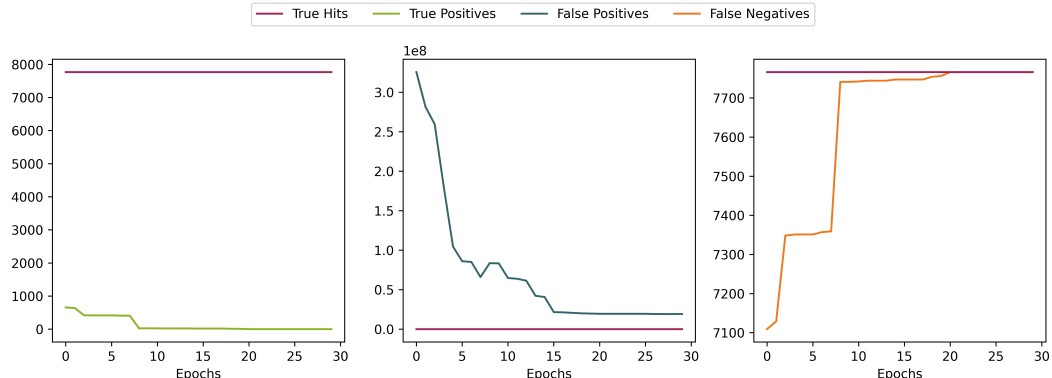

Figure 2: Comparison of evaluation metrics $TP$, $FP$ and $FN$ compared to the number of true coincidence pairs, over the course of training.

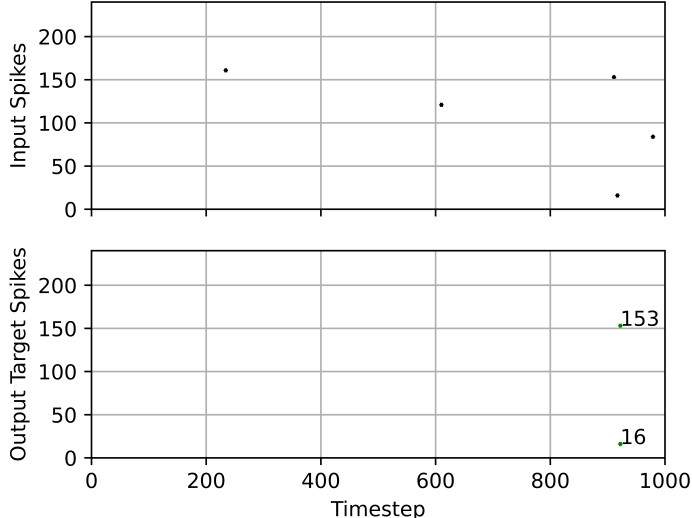

Figure 3: Visualization of the prediction task. Based on the input spike pattern (top), the SNN is trained to identify only coincidence pairs (bottom). Each timestep represents $100\,\text{ns}$, with data being generated for a simulated activity of $100\,\text{MBq}$.