# OpenReview forum: "PETNet - Coincident Particle Event Detection using Spiking Neural Networks"
_ICLR.cc/2024/Conference — ICLR 2024 Conference Withdrawn Submission_

### Official Review · Reviewer_ccA4 · 2023-10-26

**Soundness:** 2 fair
**Presentation:** 3 good
**Contribution:** 2 fair
**Rating:** 5
**Confidence:** 2

**Summary:**

This paper proposes a novel spiking neural network (SNN) architecture for coincident particle event detection in positron emission tomography (PET). Technically, the authors design a dedicated multi-objective loss function for SNNs that is both sensitive to spike counts and timing critical. Meanwhile, they implement large-scale data-parallel SNN training on a multi-node GPU system. Experiments show that PETNet outperforms SOTA algorithms with faster inference speed.

**Strengths:**

i) The topic of particle event detection using SNNs is very interesting and attractive.

ii) The authors verify a surprising conclusion that SNNs can speed up this task and even improve detection accuracy by learning coincidence patterns.

iii) The writing is straightforward, clear, and easy to understand.

**Weaknesses:**

i) I am curious and surprised by a sentence “SNN provide a promising alternative with the potential of surpassing conventional ANN prediction accuracy while requiring substantially less computational resources” in the introduction section. Why can SNNs have better performance than the corresponding ANN model and reduce computational complexity? I am curious and surprised by a sentence made in your introduction. Why can SNNs have better performance than the corresponding ANN model and reduce computational complexity? I am curious and surprised by a sentence made in your introduction. Why can SNNs have better performance than the corresponding ANN model and reduce computational complexity? Please prove this point in terms of theoretical explainability and experimentation. To the best of my knowledge, one of the biggest advantages of SNNs over ANNs is low power consumption.

ii) The authors should show more visualization results for better understand the task of particle event detection.

iii) The authors should give the detailed SNN architecture in the manuscript.

**Questions:**

See weakness.

---

> ### Author Response · Authors · 2023-11-17
> **Response to Reviewer ccA4:**
>
> As stated in the general comment, it has been shown in literature [1] that is possible for an SNN to deliver comparable, if not better, performance than an ANN. This is mainly attributed to the sparse, binary, time-resolved nature of the data.
>
> With regards to the SNN model architecture, we would like to point the reviewer to page 5, at the start of section 4, where we mention the exact design of the SNN model, with the exact numbers mentioned in Table 1. It consists of a fully connected SNN with only a single hidden layer in any case. Depending on the dataset used (Clinical, SAFIR, with/without geometry) the number of input and output nodes varied depending on the dataset, while the number of hidden layer nodes was adapted only on whether we investigated the ‘clinical’ or ‘SAFIR’ dataset. We wished to include further visualization, for example a figure visualizing the SNN architecture, but were unable to do so due to the page-limit constraint.
>
> References:
>
> [1] Deng, L., Wu, Y., Hu, X., Liang, L., Ding, Y., Li, G., ... & Xie, Y. (2020). Rethinking the performance comparison between SNNS and ANNS. Neural networks, 121, 294-307.

---

> > ### Comment · Reviewer_ccA4 · 2023-11-21
> > **Thanks for the author's response, I stick to the original score.**
> >
> > Regarding the literature you cited (Deng et al., Neural Networks, 2020), I think that the conclusions may not be completely credible, and I suggest reconsidering them from an experimental standpoint at the very least.
> > In addition, the authors should show more visualization results for better understand the task of particle event detection in future version.

---

> > > ### Author Response · Authors · 2023-11-22
> > >
> > > Thank you for your consideration. The supplementary material we added also contains a sample visualization of the dataset.

---

### Official Review · Reviewer_iojo · 2023-10-30

**Soundness:** 3 good
**Presentation:** 3 good
**Contribution:** 3 good
**Rating:** 5
**Confidence:** 4

**Summary:**

This article formulates the type II coincidence pairing problem from PET dataset and highlights the time reduction and accuracy improvement over classical SCW algorithm using a well-designed SNN training method (PETNET).

**Strengths:**

please see Summary

**Weaknesses:**

1. The motivation of this research has migrated SNN into a new medical classification problem (i.e. PET data). The writer introduce a multi-objective loss function where data sparsity and temporal resolution are both considered. The PET dataset characteristics may well capture SNN inherent spatiotemporal design that could make up good performance. However, in most research topics, ANN is always outperforming SNN, especially in terms of precision (e.g. CIFAR-10, imageNet). It is questionable for directly comparison only between non-ML algorithm (SCW) and SNN.

2. The multi-node GPU contributions is migrating the classification problem from non-GPU framework to GPU-accelerated framework (i.e. cuda). This acceleration raised the idea that if the problem can be fulfilled using only very simple ML algorithm that also can process in a very fast and accurate manner, such as SVM, decision trees or clustering if considering only single fully connected layer is needed for the SNN.

3. It is suggested to combine the Figure 1 figure 2 into one figure. Especially the PET dataset, as a subset of biomedical signal, is little covered by research topic. It is at best to illustrate the dataset in picture and visualise 1-2 selected examples from numerical vector/matrix in your dataset.

4. Extending from (1), it is also suggested to include state-of-the-art algorithm which also capture temporal dynamics apart from single-layer LSTM. For example, transformer type model vs SNN approach or CNN based classification models. The expected SNN result may also be deployed on neuromorphic hardware instead of parallel hybrid “supercomputer”. If supercomputer is available, computing resources can be sacrificed for shorter time as there could be many available ML choices apart from SNN.

**Questions:**

please see the weakness

---

> ### Author Response · Authors · 2023-11-17
> **Response to Reviewer iojo:**
>
> We do believe that the comparison against a state-of-the-art non-ML algorithm baseline is viable due to our desire to reduce computational times with the SNN. Similar surrogate models have recently proven their worth, e.g., in climate simulations [2]. If the reviewer can provide us with literature on classical machine learning algorithms that are tailored to sparse, binary, time- and spatially-resolved data, we would highly appreciate the effort.
>
> The usage of the hybrid supercomputer for data-parallel training on multiple GPUs was necessary due to the inefficient implementation of the SNN in a dense tensor framework, as stated in the conclusion section. We are currently planning future work for a more efficient implementation in this regard. However, the main purpose of this study was to showcase the adequacy of SNNs for the prediction task at hand, and the potential in saving computation time at inference.
>
> We decided against the inclusion of a visualization of the dataset due to the sparse nature of the data and therefore its lack of visible features. Please find an example in the attached supplementary material.  Figure 2 was meant to serve this purpose, detailing a schematic representation of a sample of input data and output labels. The utilization of PETNet on neuromorphic hardware is an intriguing idea for future work. As of now we unfortunately do not have access to such devices at the time of writing and thus were unable to include it.
>
> References:
>
> [2] Lam, R., Sanchez-Gonzalez, A., Willson, M., Wirnsberger, P., Fortunato, M., Pritzel, A., ... & Battaglia, P. (2022). GraphCast: Learning skillful medium-range global weather forecasting. arXiv preprint arXiv:2212.12794.

---

### Official Review · Reviewer_xnxs · 2023-10-31

**Soundness:** 3 good
**Presentation:** 2 fair
**Contribution:** 2 fair
**Rating:** 3
**Confidence:** 5

**Summary:**

Spiking Neural Networks (SNNs) are explored as an energy-efficient alternative for processing Positron Emission Tomography (PET) data. The study introduces PETNet, a method that uses SNNs to filter invalid photon hits in PET imaging. Results show PETNet has a 95.2% F1 score and is 36 times faster than traditional methods.

**Strengths:**

The study utilizes Spiking Neural Networks (SNNs) to develop PETNet, which not only achieves an impressive F1 score of 95.2% in photon coincidence detection but also processes data at a remarkable speed.

**Weaknesses:**

While the author applied SNN technology to PET data processing, the innovative aspects of the research still appear limited. The LSTM mentioned in the article fails to effectively capture long-range dependencies, and the transformer faces challenges related to memory consumption due to sequence length. Regrettably, the author did not compare with these methods in the experiments.

**Questions:**

The SNN using LIF neurons primarily relies on the constant between membrane potentials to determine its temporal dependency, which may be inadequate. Moreover, it adopts the standard fully-connected SNN design. Compared to architectures like LSTM, RNN, and Transformer, where does the SNN's advantage lie? Can this be demonstrated through experimental results?

---

> ### Author Response · Authors · 2023-11-17
> **Response to Reviewer xnxs:**
>
> The advantage of the SNN lies in their nature to model binary spike trains, which fit perfectly to our problem statement. As stated in the general comment, RNN architectures like the LSTM are simply not capable of that, resulting in their inability to produce meaningful output (see supplementary results attached to this response).

---

### Author Response · Authors · 2023-11-17

We thank all reviewers for taking the time and effort to evaluate our paper. We wish to address the raised weaknesses and questions below, hoping that it will convince the reviewers about the impact or our work and the relevance to the broad ICLR community and novel machine learning applications.

Reviewers xnxs, iojo and ccA4 have correctly pointed out that our article does little to compare the PETNet SNN to other methods such as LSTM’s or Transformers, with Reviewer iojo correctly highlighting that in most research topics ANNs outperform SNNs, and Reviewer ccA4 accentuating that one of SNN’s biggest advantages lies in comparably low power consumption. However, we would like to point out, that the prediction tasks mentioned by the reviewer iojo (CIFAR-10, ImageNet) have already been solved with conventional ANN architectures, and the purpose of SNN application there was simply to match this baseline accuracy. In our case, there is no prior ML-based solution of the prediction task.
The aim of this article is to showcase a unique situation in which SNN’s are in fact more suitable, namely when processing sparse, binary, time-variant data as is common in particle detection applications. It has been shown in literature [1] that it is possible for an SNN to deliver comparable, if not better, performance than an ANN under these conditions.

Training a regular ANN, i.e., a multi-layer perceptron, on individual time-frames is infeasible due to the time-resolved nature of the data and the possibility of coincidence pairs being detected with a timing delay. On the other hand, processing the entire time series as input to an MLP is prohibitive due to the memory constraints arising. Already for the ‘clinical’ dataset of 240 input crystals and 1,000 timesteps a fully connected ANN with a single layer would have 240,000 input nodes. For a hidden layer of comparable size (such that it does not comprise a significant bottleneck), this would imply over 57 billion parameters, going well beyond GPU memory capabilities. This is exasperated when considering a larger PET detector as well as continuous application, where the number of timesteps massively increases.

As stated in Sec. 4.5., we attempted a comparison with an LSTM. Due to its poor performance and pages limitations, the details of this comparison have not been included in the initial submission. Yet, we are attaching our results here, showing that even though training and validation loss of the model converge, the prediction accuracy stagnates at 0.0 as the predicted output values tend towards zero (cf. Supplementary Materials Fig. 2). We acknowledge the reviewers’ suggestion to include these findings in the manuscript conditional acceptance.

We deem a Transformer model inadequate for the desired usage scenario for two reasons. Firstly, as stated previously, it would go well beyond available memory since the attention mechanism scales quadratically in the sequence length. For the considered data-sets at a sequence lengths of 2,000 time-steps, this would imply calculating 4,000,000 entries of the attention matrix per sample.
Secondly, it is questionable how well the attention mechanism can cope with the sparse, binary nature of the input data, ultimately relying purely on the embedding layers. An additional aspect is that at the current stage, Transformers are likely not well equipped for inference i.e., on low-powered devices with little computational power.

References:

[1] Deng, L., Wu, Y., Hu, X., Liang, L., Ding, Y., Li, G., ... & Xie, Y. (2020). Rethinking the performance comparison between SNNS and ANNS. Neural networks, 121, 294-307.